# Comparative Study of Eclipse and RayStation Multi-Criteria Optimization-Based Prostate Radiotherapy Treatment Planning Quality

**DOI:** 10.3390/diagnostics14050465

**Published:** 2024-02-20

**Authors:** John Y. K. Wong, Vincent W. S. Leung, Rico H. M. Hung, Curtise K. C. Ng

**Affiliations:** 1Department of Health Technology and Informatics, Faculty of Health and Social Sciences, The Hong Kong Polytechnic University, Hong Kong SAR, China; wongyukkwong2013@gmail.com; 2Department of Clinical Oncology, Pamela Youde Nethersole Eastern Hospital, Hong Kong SAR, China; 3Curtin Medical School, Curtin University, GPO Box U1987, Perth, WA 6845, Australia; curtise.ng@curtin.edu.au; 4Curtin Health Innovation Research Institute (CHIRI), Faculty of Health Sciences, Curtin University, GPO Box U1987, Perth, WA 6845, Australia

**Keywords:** cancer, Gleason score, organs at risk, Pareto front, Pareto surface, prostate-specific antigen, radiation therapy, target volume, toxicity, trade-off

## Abstract

Multi-criteria optimization (MCO) function has been available on commercial radiotherapy (RT) treatment planning systems to improve plan quality; however, no study has compared Eclipse and RayStation MCO functions for prostate RT planning. The purpose of this study was to compare prostate RT MCO plan qualities in terms of discrepancies between Pareto optimal and final deliverable plans, and dosimetric impact of final deliverable plans. In total, 25 computed tomography datasets of prostate cancer patients were used for Eclipse (version 16.1) and RayStation (version 12A) MCO-based plannings with doses received by 98% of planning target volume having 76 Gy prescription (PTV_76_
*D*_98%_) and 50% of rectum (rectum *D*_50%_) selected as trade-off criteria. Pareto optimal and final deliverable plan discrepancies were determined based on PTV_76_
*D*_98%_ and rectum *D*_50%_ percentage differences. Their final deliverable plans were compared in terms of doses received by PTV_76_ and other structures including rectum, and PTV_76_ homogeneity index (*HI*) and conformity index (*CI*), using a *t*-test. Both systems showed discrepancies between Pareto optimal and final deliverable plans (Eclipse: −0.89% (PTV_76_
*D*_98%_) and −2.49% (Rectum *D*_50%_); RayStation: 3.56% (PTV_76_
*D*_98%_) and −1.96% (Rectum *D*_50%_)). Statistically significantly different average values of PTV_76_
*D*_98%,_
*HI* and *CI*, and mean dose received by rectum (Eclipse: 76.07 Gy, 0.06, 1.05 and 39.36 Gy; RayStation: 70.43 Gy, 0.11, 0.87 and 51.65 Gy) are noted, respectively (*p* < 0.001). Eclipse MCO-based prostate RT plan quality appears better than that of RayStation.

## 1. Introduction

Prostate cancer ranks as the third most prevalent cancer as per the Global Cancer Statistics 2020 report [1]. In the United States (US) alone, 288,300 men were diagnosed with prostate cancer in 2023, and the disease claimed the lives of 34,700 individuals [2]. Recent studies have indicated that prostate cancer treatments, namely radiotherapy (RT) and prostatectomy, achieve comparable clinical outcomes, highlighting the significant role of RT in managing this disease [3,4].

Although prostate RT is a relatively less invasive treatment, use of ionizing radiation in this treatment may result in adverse effects on the rectum such as pain, bleeding and increase of stool frequency [5,6,7]. RT planning plays a crucial role in minimizing the dose received by nearby healthy structures known as organs at risk (OARs), and hence the adverse effects, while accurately delivering the prescribed radiation amount to the tumor (planning target volume (PTV)) for effective treatment [8,9,10]. However, the rectum being in close proximity to the PTV, along with their overlapping regions, presents a unique challenge in prostate RT planning [11,12,13]. Sometimes, clinicians prefer reducing the dose given to the rectum to mitigate the risk of rectal toxicity at the expense of dose reduction to the PTV within an acceptable range, resulting in a decrease of treatment effectiveness [14,15]. Achieving the right balance between risk mitigation and effective treatment outcomes for each patient requires an iterative process of parameter tuning and multiple re-optimizations, which can be highly time-consuming when performed by a RT treatment planner manually [16,17,18,19,20,21,22].

To address this time-consuming issue and improve RT plan quality, multi-criteria optimization (MCO) has become available in commercial RT treatment planning systems such as RaySearch Laboratories AB RayStation (Stockholm, Sweden) [17,18,20,22,23,24,25,26,27], and Varian Medical Systems, Inc. Eclipse (Palo Alto, CA, USA) over the last decade [16,17,20,21,24,28]. This function allows the planner to use an intuitive interface to explore a series of Pareto optimal plans with different dosimetric trade-offs between PTV coverage and OARs sparing through a Pareto surface (also known as Pareto front). All Pareto optimal plans on the Pareto surface are automatically computed by the planning system based on objectives set by the planner. A treatment plan is considered Pareto optimal when any improvement in one objective can only be achieved with another scarified [16,17,18,19,22].

A few studies have specifically evaluated the use of the MCO function of RayStation [23,25,26] and Eclipse [17] treatment planning systems for prostate RT planning. In 2014, Ghandour et al. [26] conducted the first clinical study to evaluate the RayStation MCO function with nine prostate cancer patients and reported that the MCO plan quality was comparable to the traditional approach but with the added benefit of planning time reduction. Similar findings of comparable plan qualities with and without using the RayStation MCO were noted in Müller et al.’s [25] study with a sample size of 10 patients in 2017, despite the reported planning time saving benefit only being an estimation due to their retrospective study design. Nonetheless, conflicting findings were reported by Kyroudi et al. [23] in 2016 that the RayStation MCO function was unreliable as there were discrepancies between Pareto optimal and final deliverable plans for their five prostate cancer cases. They expected that this discrepancy issue would be addressed by later versions of RayStation. In 2021, Park et al. [17] carried out the first study evaluating the Eclipse MCO function (which has only become available from 2017) for the prostate RT planning for 25 patients and showed that the MCO approach allowed better OARs sparing without sacrificing the PTV coverage. However, they suggested that further studies comparing MCO functions of different treatment planning systems should be conducted.

The purpose of this study was to compare the prostate RT MCO plan qualities of Eclipse version 16.1 (released in 2021) and RayStation version 12A (available from 2022) treatment planning systems in terms of discrepancies between the Pareto optimal and final deliverable plans, and dosimetric impact of the final deliverable plans generated from the Pareto optimal plans. It was hypothesized that the Eclipse MCO function outperformed that of RayStation. Our study was the first of its kind to directly compare the Eclipse and RayStation MCO-based prostate RT treatment plan qualities. Although both systems are widely used in clinical settings, there is a gap in the literature regarding their comparative performance, particularly in the context of prostate RT. This study was conducted to fill this gap and provide the clinical community with some insights on the use of MCO-based planning [17,23,25,26].

## 2. Materials and Methods

### 2.1. Patient Selection and Simulation

This was a retrospective study involving 25 prostate cancer patients at Pamela Youde Nethersole Eastern Hospital in Hong Kong Special Administrative Region, with methods based on the similar studies on evaluation of Eclipse/RayStation MCO function for prostate RT planning [17,23,25,26]. The patients were treated with volumetric modulated arc therapy (VMAT) for prostate cancer between January 2021 and December 2022. Although use of stereotactic body radiotherapy (SBRT) has increased over recent years, according to a recent study about US national trends in prostate cancer RT fractionation regimens with 302,035 patients, more than 80% of the prostate cancer patients still received conventional fractionation RT in late 2020. This study focused on VMAT rather than SBRT for increasing its relevance to the wider clinical community [29]. Patient inclusion criteria were: (1) prostate VMAT received; and (2) computed tomography (CT) simulation performed [17,23,26]. All patients received a RT dose of 76 Gy in 38 fractions. There were several patient exclusion criteria to ensure the dataset’s integrity, including: (1) patients who underwent prostate surgery; (2) incomplete medical records; and (3) inadequate imaging data for VMAT planning. The patient characteristics are shown in Table 1.

CT Big Bore (Koninklijke Philips N. V., Amsterdam, The Netherlands) was used for simulation CT scans. To achieve similar full bladder status, the patients were required to empty their bladders and then drink 400 cc of water one hour before the scans. They were all positioned in a supine position on a vacuum bag with both hands on chest for non-contrast CT scans with 120 kV, 350–450 mAs, 3 mm slice thickness, 60 cm field of view, 512 × 512 matrix size, 1.18 pixel spacing, and a standard convolutional kernel for image reconstruction as per routine protocol at Pamela Youde Nethersole Eastern Hospital [7]. This study was conducted in accordance with the Declaration of Helsinki and approved by the Institutional Review Board of Hong Kong Polytechnic University (approval number: HSEARS20230727002 and date of approval: 3 August 2023) and Research Ethics Committee of Hong Kong East Cluster of Hospital Authority of Government of Hong Kong Special Administrative Region (approval number: HKECREC-2023-009 and date of approval: 14 March 2023), and patient consent was waived due to the study’s retrospective nature.

### 2.2. Target Volumes and OARs Segmentation

All (25) CT datasets of the selected patients acquired from the CT Big Bore simulator in Digital Imaging and Communications in Medicine (DICOM) format were imported into the Eclipse version 16.1 treatment planning system for clinical target volume (CTV), PTV and OARs segmentation. The OARs (bladder, femoral head, rectum and small bowel) were manually contoured by a radiation therapist with more than 15 years of experience [7,8,9]. The CTV and PTV (prostate and seminal vesicles) were manually segmented by a radiation oncologist experienced in prostate cancer RT, based on European Society for Therapeutic Radiology and Oncology (ESTRO) consensus guidelines [30]. Margins of approximately 10 mm for the corpus of prostate and 15 mm for the seminal vesicles were used to expand the CTV as the PTV [17,25,26]. The OARs, CTV and PTV were subsequently reviewed and approved by another radiation oncologist with associate consultant grade or above for clinical use previously. These arrangements enabled the OARs and target volumes segmentation being more standardized [7,8,9].

### 2.3. MCO-Based Treatment Planning

Eclipse version 16.1 and RayStation version 12A were used for MCO-based treatment planning for each included CT dataset with TrueBeam linear accelerator and Millennium multi-leave collimator (Varian Medical Systems, Inc., Palo Alto, CA, USA), 2 full arcs and beam energy of 6 MV photons selected [31,32]. The first and second arcs rotated from 175 to 185 degrees counterclockwise, and from 185 to 175 degrees clockwise, respectively. Table 2 shows the dose/volume treatment objectives used for the MCO-based planning of the two systems as per in-house protocol [17,33]. Rectum *D*_50%_ (V_50_ ≤ 50%) and PTV_76_
*D*_98%_ were selected as trade-off criteria because of their conflicting natures [17,23], i.e., when the dose received by at least 50% of the rectal volume decreased for minimizing the adverse effects of prostate RT, the effectiveness of the prostate cancer treatment also decreased as a result of the reduction of dose received by 98% volume of PTV_76_, and vice versa [31,32].

Although both treatment planning systems allowed navigation of Pareto optimal plans with different dosimetric trade-offs between the PTV coverage and OARs sparing through their intuitive interfaces, the Eclipse system required prerequisite plan generation based on its inverse planning method (Photon Optimization Algorithm version 16.1) prior to Pareto optimal plan navigation. The optimization process was automatically terminated when the objective functions converged, resulting in variable numbers of iterations across different datasets [17,34]. However, this optimization process (prerequisite plan generation) was not required for RayStation MCO-based planning.

For every dataset, five Pareto optimal plans on the Pareto surface of each system were selected for plan quality evaluation. These included two extreme plans with one having the best PTV dose and worst rectum dose, and another having the worst PTV dose and best rectum dose; and three other intermediate plans between the two extremes on the Pareto front [17,23]. Final deliverable plans were generated based on routine dose calculation algorithms (Eclipse Acuros XB model and RayStation Collapsed Cone-based dose engine) for all selected Pareto optimal plans (also known as navigated plans) [33,35]. This resulted in a total of 125 Pareto optimal and final deliverable plan pairs for each system (5 plan pairs per dataset × 25 datasets). Hence, the MCO-based treatment planning processes of the two systems were identical except for their two intrinsic differences (prerequisite plan generation required by Eclipse and their dose calculation algorithms) to achieve nearly equal conditions for their plan quality comparison.

### 2.4. Evaluation of MCO-Based Treatment Planning Quality

Discrepancies between the Pareto optimal and final deliverable plans were determined based on percentage differences of the PTV_76_
*D*_98%_ and rectum *D*_50%_. Furthermore, the final deliverable plans generated from the middle Pareto optimal plans (also known as nominal plans) on the Pareto surfaces by the two treatment planning systems were compared in terms of the doses received by the structures listed in Table 2, and homogeneity index (*HI*) and conformity index (*CI*) for the PTV_76_. *HI* and *CI* are indicators of *PTV* dose distribution uniformity and coverage, and were calculated using Equations (1) and (2) [17,36]. Figure 1 summarizes the overall study design.
(1)$$HI=\frac{\left(D_{2\%}-D_{98\%}\right)}{D_{50\%}}$$
where *D*_2%_, *D*_98%_ and *D*_50%_ are doses received by 2%, 98% and 50% of *PTV*, respectively. A *HI* value of 0 represents an absolute homogenous dose distribution which is the best.
(2)$$CI=\frac{V_{95\%}}{PTV}$$
where *V*_95%_ represents size of *PTV* receiving 95% of prescribed dose, and *PTV* is total size of this structure. A greater *CI* value indicates better *PTV* coverage.

### 2.5. Statistical Analysis

Statistical analysis was performed with use of SPSS Statistics 28 (International Business Machines Corporation, Armonk, NY, USA). Mean and standard deviations were calculated for: (1) the percentage differences of the PTV_76_
*D*_98%_ and rectum *D*_50%_ to indicate the discrepancies between the Pareto optimal and final deliverable plans; and (2) the doses received by structures listed in Table 2, *HI* and *CI* of the 25 selected final deliverable plans. A paired sample *t*-test was employed to compare these mean values of the two treatment planning systems with a *p*-value less than 0.05 representing statistically significant difference [8,9,17,37,38].

## 3. Results

Discrepancies between the Pareto optimal and final deliverable plans are noted for both Eclipse and RayStation treatment planning systems (Table 3). However, on average, the RayStation final deliverable plans showed improvements for both PTV_76_
*D*_98%_ (increased by 3.56% resulting in greater effectiveness) and dose to the OAR, rectum (decreased by 1.96% leading to adverse effect reduction) when compared with its Pareto optimal plans. In contrast, the Eclipse final deliverable plans only demonstrated an improvement for rectum sparing but the improvement extent was statistically significantly greater than that of RayStation (*p* = 0.0136).

Figure 2 shows an example of Pareto fronts of Eclipse and RayStation treatment planning systems for one included case to illustrate the dosimetric trade-offs between the PTV_76_
*D*_98%_ and rectum *D*_50%_ of the Pareto optimal and final deliverable plans. Again, RayStation demonstrated the improvements of PTV_76_
*D*_98%_ and rectum *D*_50%_ for its final deliverable plans but these plans also showed notable issues of PTV_76_
*D*_98%_ under-dosage (<76 Gy) and higher dose to rectum when compared with those of Eclipse.

Table 4 illustrates the dosimetric impact of Eclipse and RayStation final deliverable plans generated from their nominal plans. The Eclipse MCO function outperformed that of RayStation as evidenced by Eclipse average values of D_mean_ and *D*_98%_ ≥ 76 Gy, smaller mean *HI* and greater average *CI* for PTV_76_, and smaller average rectum D_mean_, V_50_ and V_20_ with statistically significantly differences for all of these metrics except mean rectum V_20_ (*p* < 0.001). However, RayStation demonstrated better sparing the other OARs (those not selected as trade-off criteria) in terms of smaller average values of bladder D_max_, V_70_ and V_55_, small bowel D_max_, and left and right femoral heads D_mean_ with all but one having statistically significantly differences (*p* < 0.001–0.032).

## 4. Discussion

To our best knowledge, this is the first study to compare the prostate RT MCO plan qualities of Eclipse version 16.1 and RayStation version 12A treatment planning systems. Hence, it advances the knowledge from other similar studies covering Eclipse version 15.6 and RayStation version 4 [17,23,25,26]. Our findings show that Eclipse outperformed RayStation in the MCO-based prostate RT treatment planning. This is within our expectation because previous studies on the RayStation MCO-based prostate RT treatment planning only demonstrated MCO plan qualities comparable to those of the conventional planning approach [25,26], or sometimes even worse [23]. However, Park et al.’s [17] study with the greatest sample size of 25 prostate cancer patients showed that the Eclipse MCO enabled better OARs sparing without sacrificing the PTV coverage. Our results of Eclipse average PTV_76_ D_mean_ and *D*_98%_ values greater than 76 Gy with mean *HI* very close to 0 (indicating homogenous dose distribution) and average CI above 1 (good PTV coverage), and smaller rectum average D_mean_, V_50_ and V_20_ are consistent with Park et al.’s findings.

However, the better performance of the Eclipse MCO function comes at a price which is delivery of higher doses to other OARs not selected as the trade-off criteria such as bladder, small bowel and femoral heads (Table 4). Similar trade-offs are also noted in Park et al.’s [17] study. Moreover, one additional process (prerequisite plan generation) is required for using the Eclipse MCO function which might have an impact on the extent of its potential planning time saving despite the fact that evaluation of planning time reduction is not within the scope of this study [17].

Although our results demonstrate that the lower performance of RayStation MCO function, clinical acceptability of its MCO plans should be determined. Given that clinical acceptable criterion for PTV under-dosage is up to 5% and its mean value of PTV_76_
*D*_98%_ was 70.43 Gy (7.3% under-dosage), its MCO plan quality appears not clinically acceptable (Table 4) [23]. Nonetheless, in order to conduct a systematic comparison of the Eclipse and RayStation MCO functions in this study, the dosimetric impacts were evaluated based on their final deliverable plans generated from the nominal plans between the two extreme Pareto optimal plans on the Pareto surfaces. A closer look at Figure 2 reveals that the above determination of clinical acceptability of RayStation MCO function appears to be oversimplified.

Clinically, radiation oncologists are usually involved in selecting a Pareto optimal plan from the Pareto front for generating the final deliverable plan [25]. Figure 2 shows that the PTV_76_
*D*_98%_ of RayStation Pareto optimal plans ranged between 70.63 and 73.92 Gy (2.7–7.1% under-dosage). However, Table 3 illustrates that the RayStation final deliverable plans generated from the Pareto optimal plans had 3.56% PTV_76_ D_98%_ increase on average. For example, Figure 2 demonstrates that the PTV_76_
*D*_98%_ range of the RayStation final deliverable plans was 72.18–75.24 Gy which only has a maximum of 5% under-dosage and should be deemed clinically acceptable. Hence, radiation oncologists need to be aware that some RayStation Pareto optimal plans on the Pareto front could have the PTV under-dosage issue, and exercise sound clinical judgement to select appropriate Pareto optimal plans for clinically acceptable practice. They should only consider the RayStation MCO function as a time-saving tool for planning primarily [23,25].

The gastrointestinal toxicity of prostate RT is a well-known issue [7]. Although the Eclipse MCO function seems effective to address this issue, the associated trade-off, increase dose to bladder appears concerning because this can cause hematuria, urinary incontinence and painful urination after prostate RT treatment [17,41]. According to a recent systematic review and meta-analysis, patients who receive the prostate RT have 21.9% and 31.9% chances of having acute gastrointestinal and genitourinary toxicities, and 16.2% and 28.0% for the late toxicities, respectively [42]. In our study, the OAR, bladder was not selected as the trade-off criterion according to similar studies on evaluation of MCO function for the prostate RT planning [23,25,39]. This arrangement facilitated better illustration of the trade-offs between the PTV and rectum doses on the two-dimensional Pareto surface. Additionally, the bladder was already included as one of the dose/volume treatment objectives and hence its selection would have a minimal effect on further optimizing the dose to it without sacrificing the others, i.e., PTV and rectum doses [23].

Use of artificial intelligence (AI) such as machine learning (ML) and deep learning has become popular in medical imaging [7,9,43,44,45]. Apart from the MCO, knowledge-based planning function based on ML technology (known as RapidPlan) has also become available in recent versions of Eclipse [46]. Some studies have already explored integrated use of Eclipse RapidPlan and MCO functions for prostate [16,28], lung [19], brain [47], and head and neck RT [21], and reported that the integrated use increased the RT plan quality in terms of better OARs sparing without sacrificing PTV coverage. Nonetheless, Jayarathna et al.’s study [16] showed that quality of their prostate RT plans generated through the combined use of these functions was only comparable to those with the use of MCO alone. This might be due to the quality of their historical plans used to train the knowledge-based planning model for optimization objective determination [19]. Unfortunately, RayStation version 12A does not allow this integrated use despite the ML knowledge-based planning function being available [35,48]. Hence, only the Eclipse and RayStation MCO functions were compared in this study.

Apart from the aforementioned issue of selection of only two trade-off criteria for the MCO-based planning, there are other limitations in this study. For example, it is a retrospective study with a relatively small sample size of 25 patients from one single center. However, our sample size was the largest among the similar studies which also collected data from a single center [17,23,25,26]. Although a prospective study would allow the evaluation of potential MCO-based planning time saving, the scope of our study was the MCO-based prostate RT planning quality in line with the previous studies [17,23]. The MCO time-saving benefits of individual systems have been reported in other studies [16,25]. Müller et al. [25] and Jayarathna et al. [16] demonstrated that the MCO functions of RayStation version 4.0 and Eclipse version 15.5 were able to reduce the prostate RT planning time by 64% and 32% (average planning time with and without MCO for RayStation (20 and 55 min) and Eclipse (28 and 41 min)), respectively. The longer time required for Eclipse MCO-based planning could be attributed to the additional prerequisite plan generation.

Furthermore, the two RT treatment planning systems provided different dose calculation algorithms (Eclipse Acuros XB model and RayStation Collapsed Cone-based dose engine) for the final deliverable plan generation, which is an unavoidable system limitation despite all other settings for the MCO-based planning being identical. Additionally, the extent of impact of different dose calculation algorithms on the final deliverable plan quality was investigated through evaluating the discrepancies between the Pareto optimal and final deliverable plans. This can be considered a merit of our study [23]. However, direct comparison of the quality of plans with and without MCO use was not performed for individual systems in this study because these have been conducted in previous studies and their findings are available elsewhere [17,23,25,26]. The purpose and novelty of this study was to compare the MCO plan qualities of the two treatment planning systems directly.

Future studies on other cancer types such as lung, brain, and head and neck cancers with prospective data collection from various centers covering multiple RT fractionation regimens such as VMAT and SBRT, greater sample sizes and use of more trade-off criteria are encouraged. Additionally, the comparison of integrated uses of knowledge- and MCO-based prostate RT plannings between Eclipse and RayStation should be conducted when such use is feasible for RayStation [16,19,21,28,47].

## 5. Conclusions

This study compared the MCO-based prostate RT treatment planning qualities between Eclipse version 16.1 and RayStation version 12A systems. Its results show that both systems had discrepancies between the Pareto optimal and final deliverable plans. The RayStation final deliverable plans showed improvements for both PTV_76_
*D*_98%_ and rectum doses when compared with its Pareto optimal plans as a result of the use of dose calculation algorithm different from Eclipse. In addition, the RayStation MCO function was able to generate clinically acceptable plans. Nonetheless, the Eclipse MCO function outperformed that of RayStation, as evidenced by the Eclipse average values of D_mean_ and *D*_98%_ ≥ 76 Gy, smaller average *HI* and greater mean *CI* for the PTV_76_, and the smaller average rectum D_mean_, V_50_ and V_20_ with the statistically significantly differences for all of these metrics except the mean rectum V_20_.

Radiation oncologists need to be aware of these system specific characteristics when using MCO functions for optimizing treatment outcomes in clinical practice. Future studies on other cancer types such as lung, brain, and head and neck cancers with the prospective data collection from various centers covering multiple RT fractionation regimens such as VMAT and SBRT, greater sample sizes and use of more trade-off criteria are encouraged.

## Figures and Tables

**Figure 1 diagnostics-14-00465-f001:**
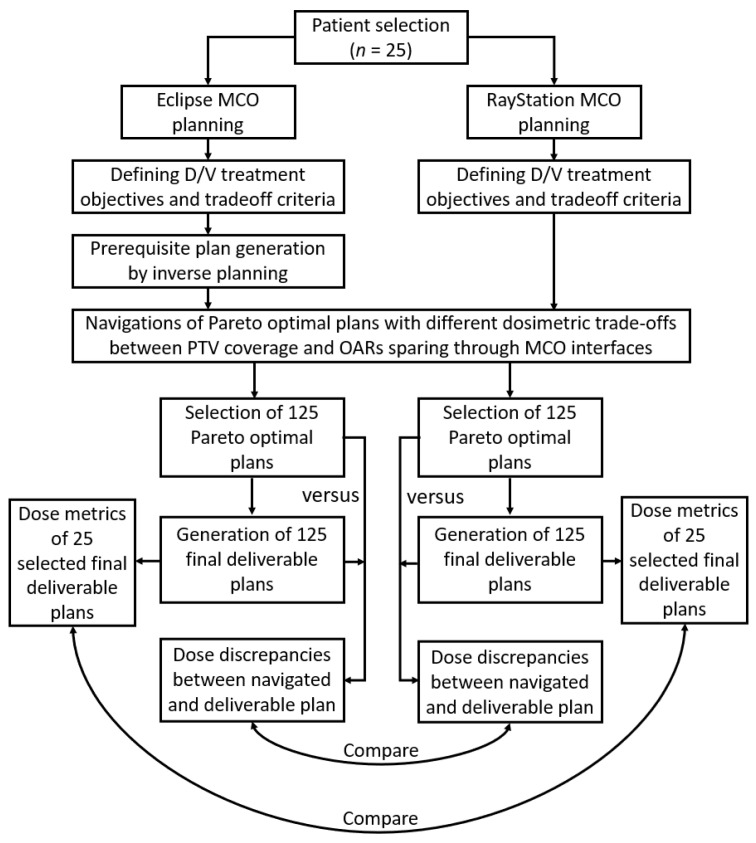
Study design overview. D/V, dose/volume; MCO, multi-criteria optimization; OARs, organs at risk; PTV, planning target volume.

**Figure 2 diagnostics-14-00465-f002:**
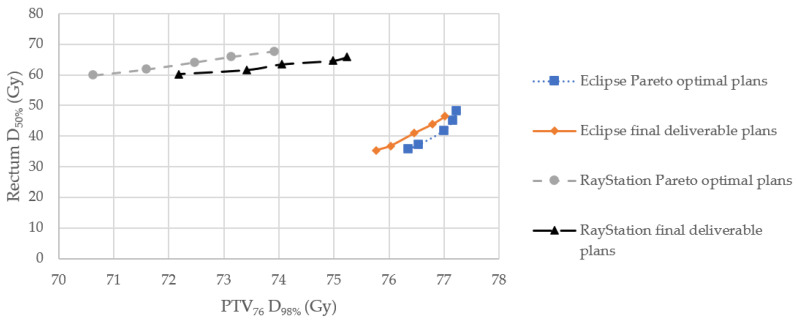
Example of Pareto fronts of Eclipse and RayStation treatment planning systems. *D*_50%/98%_, dose received by 50/98% of structure, respectively; PTV_76_, planning target volume receiving 76 Gy dose.

**Table 1 diagnostics-14-00465-t001:** Patient characteristics (*n* = 25).

Characteristics	Value
*Age*	
60–69 years	3 (12%)
70–79 years	18 (72%)
>80 years	4 (16%)
*Tumor classification*	
T1	6 (24%)
T2	8 (32%)
T3	11 (44%)
T4	0 (0%)
N0	25 (100%)
N1	0 (0%)
M0	25 (100%)
M1	0 (0%)
*Histology*	
Adenocarcinoma	6 (24%)
Acinar adenocarcinoma	8 (32%)
Unknown	11 (44%)
*Pre-treatment PSA level* (ng/mL)	
<10	12 (48%)
10–20	9 (36%)
>20	4 (16%)
*Pre-treatment GS*	
≤6	7 (28%)
7	14 (56%)
≥8	4 (16%)
*Median PTV size* (cm^3^)	98.48

Figures in parentheses are proportions. PSA, prostate-specific antigen; GS, Gleason score; PTV, planning target volume.

**Table 2 diagnostics-14-00465-t002:** Dose/volume treatment objectives for multi-criteria optimization-based prostate radiotherapy treatment planning.

Structure	Dose/Volume Objectives
PTV_76_	*D*_2%_ ≤ 105%
	*D*_99%_ ≥ 100%
	*D*_98%_ ≥ 100%
Rectum ^1^	D_max_ < 105%
	V_70_ ≤ 20%
	V_50_ ≤ 50%
Bladder ^1^	V_70_ ≤ 20%
	V_55_ ≤ 50%
Small bowel ^1^	D_max_ < 52 Gy
Femoral head ^1^	V_50_ ≤ 5%

^1^ Organs at risk. CTV_76_, clinical target volume receiving 76 Gy dose; *D*_2%/98%/99%_, dose received by 2%/98%/99% of structure; D_max_, maximum dose received by structure; PTV_76_, planning target volume receiving 76 Gy dose; V_50/55/70_, volume of structure receiving 50/55/70 Gy dose.

**Table 3 diagnostics-14-00465-t003:** PTV_76_
*D*_98%_ and rectum *D*_50%_ percentage differences between Pareto optimal and final deliverable plans.

Parameter	% Difference		*p*-Value
	Eclipse	RayStation	
PTV_76_ *D*_98%_	−0.89 ± 0.68%	3.56 ± 1.90%	<0.001
Rectum *D*_50%_	−2.49 ± 2.99%	−1.96 ± 2.59%	0.0136

Figures are expressed in mean ± standard deviation. *D*_50%/98%_, dose received by 50%/98% of structure, respectively; PTV_76_, planning target volume receiving 76 Gy dose.

**Table 4 diagnostics-14-00465-t004:** Dosimetric impact of Eclipse and RayStation final deliverable plans in multi-criteria optimization-based prostate radiotherapy treatment planning.

Structure	Parameter	Reference Constraint Value	Eclipse	RayStation	*p*-Value
PTV_76_	D_mean_ (Gy)	-	78.64 ± 0.32	76.88 ± 0.78	<0.001
	*D*_98%_ (Gy)	≥72.2 [26,39]	76.07 ± 0.67	70.43 ± 1.55	<0.001
	*HI*	-	0.06 ± 0.01	0.11 ± 0.02	<0.001
	*CI*	-	1.05 ± 0.03	0.87 ± 0.04	<0.001
Rectum	D_max_ (Gy)	-	81.25 ± 0.90	78.43 ± 0.74	<0.001
	D_mean_ (Gy)	-	39.36 ± 4.84	51.65 ± 7.51	<0.001
	V_50_ (cm^3^)	<50% [40]	23.56 ± 10.06	34.96 ± 13.39	<0.001
	V_20_ (cm^3^)	-	44.04 ± 22.59	45.29 ± 22.71	0.423
Bladder	D_max_ (Gy)	-	82.32 ± 0.52	79.14 ± 0.78	<0.001
	D_mean_ (Gy)	-	34.28 ± 9.81	34.95 ± 9.69	0.404
	V_70_ (cm^3^)	<35% [40]	61.19 ± 21.39	37.55 ± 13.43	0.001
	V_55_ (cm^3^)	-	84.28 ± 27.78	63.31 ± 20.49	0.002
Small bowel	D_max_ (Gy)	≤55 [40]	21.92 ± 21.35	19.49 ± 18.35	0.334
Left femoral head	D_mean_ (Gy)	-	16.44 ± 3.19	11.87 ± 3.96	<0.001
Right femoral head	D_mean_ (Gy)	-	17.15 ± 4.97	11.57 ± 4.02	0.032

Figures are expressed in mean ± standard deviation. *CI*, conformity index; *D*_98%_, dose received by 98% of structure; D_mean/max_, mean/maximum dose received by structure, respectively; *HI*, homogeneity index; PTV_76_, planning target volume receiving 76 Gy dose; V_20/50/55/70_, volume of structure receiving 20/50/55/70 Gy dose, respectively.

## Data Availability

The datasets used in this study are not publicly available due to strict requirements set out by the Research Ethics Committee of Hong Kong East Cluster of Hospital Authority of Government of Hong Kong Special Administrative Region.

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
