# Peer review of "Comparative Study of Eclipse and RayStation Multi-Criteria Optimization-Based Prostate Radiotherapy Treatment Planning Quality"

_diagnostics, 2024, doi:10.3390/diagnostics14050465_

Round 1

Reviewer 1 Report

Comments and Suggestions for Authors

The authors present a comparison of the MCO functions of Eclipse and RayStation for prostate radiotherapy planning regarding discrepancies between Pareto optimal and final deliverable plans, and the dosimetric impact of final deliverable plans. This study involved 25 computed tomography datasets of prostate cancer patients. PTV76 D98% and rectum D50 were used as trade-off criteria. Eclipse and RayStation systems showed discrepancies between Pareto optimal plans and final deliverable plans. Percentage differences for Eclipse were -0.89% (PTV76 D98%) and -2.49% (Rectum D50%), while for RayStation, they were 3.56% (PTV76 D98%) and -1.96% (Rectum D50%). Homogeneity Index (HI), Conformity Index (CI), and mean dose received by the rectum were noted between the two systems. Based on the obtained results, the authors suggested that the quality of prostate radiotherapy planning based on Eclipse MCO is superior compared to that based on RayStation.

The manuscript addresses a topic of clear interest regarding the multi-criteria optimization functions of Eclipse and RayStation for prostate radiotherapy. However, there are several critical points identified about the completeness and relevance of the presented results.

The first issue is the decision to study conventional fractionation for prostate treatment, considering the growing trend toward using Stereotactic Body Radiation Therapy (SBRT). The primary toxicities in such treatments are genitourinary, and focusing on rectum dose/volume values may not fully reflect current clinical challenges. An approach aligned with modern treatment trends could enhance the clinical relevance of the study. Manual segmentation of the PTV, as described in the manuscript at line 131, raises concerns about consistency and objectivity. Predetermined margins could have provided a more standardized and transparent methodology. Clinically unacceptable results obtained with RayStation raise questions about the practicality and reliability of such a system for treatment planning. The discussion should better explore the practical implications of these results and their impact on clinical decisions. The absence of a manual planning benchmark represents a methodology gap. A direct comparison with a manual treatment plan would have provided a more solid basis for evaluating the performance of Eclipse and RayStation MCO functions, highlighting real differences compared to a traditional approach. The Introduction section is very unbalanced, with much too large weight on the clinical background. The Introduction should move much quicker toward the topic of the paper.

Considering the evolution of treatment planning techniques and the increasing importance of personalized treatments, I believe that the study lacks novelty and interest for the readers in this form. Nevertheless, I appreciate the time and effort the authors have put into this research.

Reviewer 2 Report

Comments and Suggestions for Authors

This article presents a very interesting study. There are not many things to change. I only have one comment. In table 4, it would more complete to include one more column, with reference values, such as the prescribed dose,  and the dose limit for toxicity in different OARs. Excellent work.!

Reviewer 3 Report

Comments and Suggestions for Authors

The authors report a study to compare the prostate RT MCO plan qualities of two available versions of TPSs in terms of discrepancies between Pareto optimal and final deliverable plans, and dosimetric impact of final deliverable plans. In my opinion, the manuscript is reasonably twell-written. Moreover, The comparison is useful to the RT community, although not having the latest version of each software reduces its usefulness for choosing between the two TPSs. Some relatively improvements, however, are required in my opinion, as detailed below:

1. Please either discuss and convince the readers that the corresponding planning software versions used in this study are equivalent to the latest versions of the RayStation and Eclipse software for the purposes of this study, or explicitly state the software versions in the abstract.

2. Please add a more convincing argument at the end of the Introduction regrading the novelty of this work and, specifically, how it attempts to fill the gap in the current knowledge on this research question.

3. It seems necessary to add more information on the details of how each system was used and the efforts made to ensure that the two systems were compared in equal conditions. For example, please add the optimization termination criterion for each system, number of iterations, and any further details of the optimization settings. Given the retrospective nature of this study, this is an important issue.

4. It will be interesting and useful to provide some comments and discussion of the practical matters regarding the two TPSs, for instance, the calculation and optimization times (while providing the main specifications of each computer), the extra time needed to do the prerequisite plan generation in Eclipse as an additional step, etc.

5. It will also be useful to mention any other direct comparisons made between the MCO algorithms of theses two TPSs, albeit for non-prostate cases (e.g., Nguyen et al. Comparison between multi-criteria optimization (MCO) (Raystation®) and Progressive Resolution Optimizer (PRO) (Eclipse®) for the dosimetry of breast cancer with prophylactic nodal irradiation treated with volumetric modulated arc therapy (VMAT), Physica Medica, Volume 32, Supplement 4, 2016, Pages 356-357, https://doi.org/10.1016/j.ejmp.2016.11.084.)

Comments on the Quality of English Language

Generally well-written. Some paragraphs are better to be split into two or more paragraphs (e.g., lines 99 to 121).

Round 2

Reviewer 3 Report

Comments and Suggestions for Authors

I would like to thank the authors for addressing my comments to a reasonably acceptable level.